# Spineless Cactus plus Urea and Tifton-85 Hay: Maximizing the Digestible Organic Matter Intake, Ruminal Fermentation and Nitrogen Utilization of Wethers in Semi-Arid Regions

**DOI:** 10.3390/ani12030401

**Published:** 2022-02-08

**Authors:** Robert E. Mora-Luna, Ana M. Herrera-Angulo, Michelle C. B. Siqueira, Maria Gabriela da Conceição, Juana C. C. Chagas, Carolina C. F. Monteiro, Antonia S. C. Véras, Francisco F. R. Carvalho, Marcelo A. Ferreira

**Affiliations:** 1Deanship of Research, Coordination of Agricultural Research, National Experimental University of Táchira (UNET), San Cristobal 5001, Táchira, Venezuela; aherrera@unet.edu.ve; 2Department of Animal Science, Federal Rural University of Pernambuco (UFRPE), Recife 52171900, PE, Brazil; michelle.siqueira2@gmail.com (M.C.B.S.); gabizoo2283@hotmail.com (M.G.d.C.); monteirocarolinac@gmail.com (C.C.F.M.); antonia.veras@ufrpe.br (A.S.C.V.); francisco.rcarvalho@ufrpe.br (F.F.R.C.); marcelo.aferreira@ufrpe.br (M.A.F.); 3Department of Agricultural Research for Northern Sweden, Swedish University of Agricultural Sciences (SLU), 90183 Umea, Sweden; 4Department of Animal Science, Alagoas State University (UNEAL), Santana do Ipanema 57500000, AL, Brazil

**Keywords:** intake, nitrogen balance, roughage, ruminal fermentation, semi-arid

## Abstract

**Simple Summary:**

In semi-arid regions, providing a roughage adapted to water deficient conditions, such as spineless cactus, associated with a source of physically effective fiber, could be a feed alternative for sheep. Five inclusion levels of spineless cactus plus urea and ammonium sulfate to replace Tifton-85 hay were tested in sheep diets with a roughage/concentrate ratio of 70:30. The dry matter and digestible organic matter intake, as well as ruminal fermentation, nitrogen balance, and microbial protein supply, were evaluated. The results suggested that spineless cactus inclusion affected quadratically the dry matter and digestible organic matter intake, as well as retained nitrogen and microbial protein supply. The spineless cactus plus urea and ammonium sulfate improved nitrogen utilization, reducing linearly urinary nitrogen excretion, serum urea, and ammonia plasma. On the other hand, spineless cactus inclusion increased the ruminal acetate and propionate concentrations, while ruminal pH and ruminal ammonia nitrogen were decreased. We recommend a roughage consisting of spineless cactus (plus urea and ammonium sulfate) and Tifton-85 hay in a 41:29 ratio, in order to maximize the digestible organic matter intake and N-utilization. This could lead to an improvement in the productive performance of animals in semi-arid regions.

**Abstract:**

The aim of this study was to evaluate the effect of replacing Tifton-85 hay (*Cynodon* spp. cv. Tifton 85) with 0, 150, 300, 450 and 600 g/kg dry matter (DM) of spineless cactus (SC, *Nopalea cochenilifera* Salm-Dyck) plus urea and ammonium sulfate (UAS; 9:1) on DM, digestible organic matter (DOM) and indigestible neutral detergent fiber (iNDF) intakes, as well as ruminal fermentation, N-balance, and microbial protein supply (MPS). Five rumen-fistulated and cannulated crossbred wethers, weighing 43.8 ± 5.80 kg, were randomized in a 5 × 5 Latin square design. Isonitrogenous diets (14% crude protein) were supplied with a roughage/concentrate ratio of 70:30. The DOM intake, N-retained, and MPS showed quadratic responses (*p <* 0.05), with maximum values estimated at the levels of SC+UAS of 414, 438 and 418 g/kg DM, respectively. Rumen pH and ammonia nitrogen, iNDF intake, N-urinary excretion, and serum urea and plasma ammonia reduced linearly (*p <* 0.05) with increasing SC+UAS inclusion. Ruminal acetate and propionate concentrations increased linearly with increasing SC+UAS inclusion. In wethers fed diets with a roughage/concentrate ratio of 70:30, roughage constituted of a SC+UAS/hay (Tifton-85) ratio of 41:29 is recommended in order to maximize the DOM intake, N-retention, and MPS.

## 1. Introduction

Sheep production is a major economic activity in arid and semi-arid regions. Sheep can make use of low-quality biomass in times of scarcity and transform it into useful products, such as milk, meat and wool [1]. Considering the expansion of semi-arid regions [2,3], it is necessary to find a roughage source with great yield potential that is adapted to adverse edaphoclimatic effects, mostly water deficit [4]. Spineless cactus (SC) is a possible solution.

Spineless cactus is an excellent source of non-fiber carbohydrates (NFC; 52.3–66%), but concentrations of neutral detergent fiber (NDF; 21.7–27.7%) and crude protein (3.4–4.1%) [5,6,7] are insufficient to sustain desired levels of animal performance. This roughage should thus be combined with another forage that provides physically effective fiber and a protein source [8].

Conserved forages, such as silage and hay, provided in semi-arid regions, are made and marketed in other regions without hydric limitations, an aspect that promotes increased feed expenses and forage prices greater than those of SC (0.27 vs. 0.13 US$/kg dry matter [DM]) [9]. However, offering these conserved forages as sources of physically effective fiber in combination with SC could provide a feed alternative to decrease costs and provide efficient resources for semi-arid regions. Tifton-85 hay (*Cynodon* spp. cv. Tifton 85) can be used as a fiber source (NDF: 770 g/kg DM) [10]. This roughage is considered a high-quality tropical forage, due to its rapid growth rate and high in vitro DM digestibility values (592 g/kg DM) relative to other bermudagrass hybrids, such as Tifton-78, Tifton-44, and Coastal (568, 532, and 523 g/kg DM, respectively) [11].

It was hypothesized that there is a SC/Tifton-85 hay ratio that maximizes nutrient intake and benefits in sheep diets. Thus, the aim was to evaluate the effect of the inclusion of SC (*Nopalea cochenilifera* Salm-Dyck) adjusted with urea and ammonium sulfate mix (UAS) in Tifton hay-based diets on DM, digestible organic matter (DOM), and indigestible neutral detergent fiber (iNDF) intakes, ruminal fermentation, nitrogen (N) balance, and microbial protein supply (MPS).

## 2. Materials and Methods

The experiment was carried out in Animal Science Department at the Federal Rural University of Pernambuco, located in Recife, Pernambuco State, Brazil (08°01′13.4′′ S 34°57′14.9″ W), with climate classified according to Köppen-Geiger as tropical hot and humid (As’) [12]. The altitude is 4 m, with an annual precipitation averaging 1804 mm and an annual temperature averaging 27.5 °C, ranging from 24 to 31 °C.

The Tifton-85 hay was purchased from the local market in Recife, Pernambuco, Brazil, while SC was provided by the Caruaru Experimental Station of the Agronomic Institute of Pernambuco (IPA), Caruaru, Pernambuco, Brazil, situated 130 km away from Recife. The SC was harvested every 14 days, and was transported from Caruaru to Recife for storage.

The animals used in the experiment were registered and cared for in accordance with the guidelines and recommendations of the Committee of Ethics on Animal Studies at the Federal Rural University of Pernambuco (License N° 069/2016).

### 2.1. Animals, Management, and Experimental Design

Five rumen-fistulated and cannulated crossbred wethers (of no defined breed), averaging 43.8 ± 5.80 kg body weight (BW) and 14 ± 2.33 months in age, were assigned randomly into five treatments on a 5 × 5 Latin square design. The animals were treated for internal parasites before the start of the experiment and were housed in individual pens (0.93 × 1.54 m) fitted with feeders and a fresh water source. Water was offered ad libitum.

Diets were supplied ad libitum as a total mixed ration, twice per day at 8:00 and 16:00 h. The amount of feed offered to the wethers was adjusted daily to allow refusals of approximately 5–10% of total DM provided. The experiment lasted 110 days with five experimental periods of 22 days, after 14 days of diet adaptation [13], and 8 days for sampling and data recording.

The chemical composition and nutritional value of the dietary ingredients are shown in Table 1. The experimental treatments are presented in Table 2 and consisted of Tifton-85 hay as the exclusive source of roughage in the diet and replacement of 20, 40, 60 and 80% of Tifton-85 hay by SC plus UAS mixture (9:1), which represented DM inclusions of 150, 300, 450 and 600 g SC+UAS/kg DM, respectively, in the diet. The UAS mixture was added to the diets in order to promote isonitrogenous experimental diets at 14% of crude protein concentration and to meet the requirements of sheep with an average daily gain of 250 g/d [14]. The concentrates were composed of soybean meal, ground corn, common salt, and mineral mixture, with a roughage/concentrate ratio of 70:30 (Table 2).

### 2.2. Data and Sample Collection

The roughage and concentrate were offered and refusals were weighed daily (Ramuza Scale, Model Ramuzatron 15, Santana de Parnaíba, São Paulo, Brazil) from the 15th to the 22nd day of the experimental period, for estimation of the nutrient intake. Samples of feeds and refusals were pre-dried in a forced-air oven (Tecnal, Model TE 394-2, Piracicaba, São Paulo, Brazil) at 55 °C to constant weight, and one sample was prepared per animal in each period for subsequent chemical analysis.

To estimate the DOM intake, the total fecal collection was performed from the 17th to the 19th day of each experimental period, using collection bags fixed on the animals. Feces were removed from the collection bags every 6 h. Samples of each collection day were immediately pre-dried in a forced-air oven at 55 °C for 72 h. A composite sample was obtained for each animal in each period for chemical analysis.

On the 16th day of each period, samples of ruminal contents were taken at 0 h (before the morning feeding was offered) and at 2, 4, 6, and 8 h after the feed was offered. Samples were collected from the anterior dorsal, anterior ventral, medium ventral, and posterior dorsal ruminal regions. Ruminal contents were filtered through four layers of cheesecloth for immediate pH determination by using a pH meter (Kasvi, Model K39-0014P, São José do Pinhais, Paraná, Brazil). From the extracted ruminal fluid (at each pH sampling time), 30-mL aliquots were deposited in plastic vials with 1.5 mL 6 N HCl. The samples were stored at –20 °C until analysis of rumen ammonia nitrogen (RAN) and volatile fatty acids (VFAs).

To estimate N-balance and purine derivative (PD) excretion, total urine collection was performed during the three days of fecal collection. Funnels and hoses were coupled to abdominal area and penis of the animals, and urine was conducted into a container containing 100 mL of 10% sulfuric acid. The pH was measured every 6 h to keep it below 3.0 [15] to prevent both ammonia N-loss [16] and bacterial destruction of PDs [17]. At the end of each collection day, weight and total urine volume were determined, which was diluted to 4 L [16] with distilled water and filtered through four layers of cheesecloth. Aliquots of 50 mL were frozen at −20 °C for chemical analysis.

On the 19th day of each experimental period, blood samples were extracted from the animals 4 h after morning feeding by jugular puncture with 21 G × 1″ needles (Vacuette, Greiner Bio-One, Kremsmünster, Austria), using Vacuette tubes (Greiner Bio-One, Americana, São Paulo, Brazil) with and without anticoagulant (EDTA). Blood samples were centrifuged (Centribio, Model 80-2B 15 mL, Mombai, India) at 1016× *g* for 15 min, and the plasma and serum obtained stored at −20 °C.

### 2.3. Chemical Analyses

Composite samples of feed, refusals, and feces were ground in a Willey mill (Marconi, Model MA 340, Piracicaba, Brazil) in 1- and 2-mm sieves for chemical analysis. According to the AOAC [18], the samples were analyzed for DM by gravimetric estimation at 105 °C (code 930.04), for N by the Kjeldahl method (code 984.13), and for ash by ignition at 600 °C (code 942.05). All these chemical analyses were performed in samples processed in 1-mm sieve. Urine N was determined by the Kjeldahl method.

To determine iNDF concentration and iNDF intake, feed and refusal samples (processed in 2-mm sieve) were incubated in a bovine rumen for 288 h [19] for subsequent determination of NDF concentration. The NDF (ingredients, refusals, feces, rumen-incubated material) determinations were made with heat-stable amylase (Termamyl 2X, Novozymes, Copenhagen, Denmark) without sodium sulfite [20], corrected for nitrogen compounds [21], and expressed as ash-free organic matter (OM) (aNDF(n) [20]. Acid detergent fiber and lignin for all ingredients were determined according to Van Soest [22], with the exception that lignin concentration of SC was determined by potassium permanganate oxidation [23].

For rumen VFA and RAN determination, upon thawing, a 7-mL aliquot of ruminal fluid samples were centrifuged (CentriBio, Modelo 80-2B-15 mL, Mumbai, India) at 1016× *g* for 10 min and the supernatant used for VFA determination (acetate, propionate, and butyrate) by gas chromatography [24]. CG-Master (Brazil) equipment with a flame ionization detector and Carbowax column (DB-Wax; 60 m, 0.25 mm × 1 μm) was used. Vaporizer, detector, and oven temperatures were 200, 200, and 120 °C, respectively. A 2-mL aliquot of supernatant was recentrifuged at 20,817× *g* and 4 °C for 30 min (Eppendorf, Model AG2231, Hamburg, Germany) and the supernatant used for RAN determination by colorimetry [25]; the absorbance was measured on an Agilent spectrophotometer (Model 8453, Santa Clara, CA, USA).

To determine glucose (in serum), uric acid (in urine) and urea (in serum and urine), Labtest kits (Labtest Diagnostica SA, Lagoa Santa, Minas Gerais, Brazil) were used in Labmax 240 equipment (Labtest, Prestige Model 24i, Tokyo, Japan). For the determination of urine ammonia, the colorimetric procedure of Chaney and Marbach [25] was used, and an adaptation of this method was used for plasma ammonia determination [26]. Urine allantoin concentration was determined by colorimetry as described by Chen and Gomes [17], and the absorbance was measured on the spectrophotometer previously described.

### 2.4. Calculations

Chemical compositions of diets were calculated from the proportion of ingredients and their respective values. The NFC was calculated according to Detmann and Valadares Filho [27]. The OM was calculated as 1000 g/kg DM—g ash/kg DM. The DM, iNDF, and N intakes were calculated by subtracting their contents in the refusals from the daily amounts offered. Nitrogen balance estimate was obtained by subtracting urinary and fecal N-excretion from N-consumption, the latter recorded during the three days of collection of feces and urine.

The PDs excreted were calculated as the sum of daily urinary excretion of allantoin and uric acid, without considering xanthine and hypoxanthine excretion, since allantoin + uric acid is highly correlated with rumen nucleic acid concentration [28]. The PDs absorbed were calculated according to the mathematical model described by Chen et al. [15]. The microbial N-supply (MNS) was estimated according to Chen et al. [16] and the MPS was calculated as MNS × 6.25. Digestible OM intake was calculated as OM intake × OM digestibility, and the digestible OM apparently digested in the rumen (DOMR) was calculated assuming 65% of DOM intake [29].

### 2.5. Statistical Analysis

Data were analyzed by ANOVA using the MIXED procedure of SAS (Version 9.4; SAS Inst., Inc., Cary, NC, USA) according to a 5 × 5 Latin square design. The statistical model used was
*Y_ijk_* = *µ* + *D_i_* + *P_j_* + *A_k_* + *E_ijk,_*
(1)
where *Y_ijk_* is a dependent variable, *µ* is the mean for all observations, *D_i_* is the fixed effect of diet *i*, *P_j_* is the random effect of period *j*, *A_k_* is the random effect of animal *k*, and *E_ijk_* ~*N*(0, *σ*^2^*_e_*) represents the residual error.

Rumen pH, RAN, and VFA were analyzed as repeated measures data [30]. The statistical model used was
*Y_ijk_* = *µ* + *D_i_* + *P_j_* + *A_k_* + *T_l_* + (*D* × *T*)*_il_* + *E_ijk,_*
(2)
where *Y_ijk_* is a dependent variable, *µ* is the mean for all observations, *D_i_* is the fixed effect of diet *i*, *P_j_* is the random effect of period *j*, *A_k_* is the random effect of animal *k*, *T_l_* is the fixed effect of collection time *l*, (*D* × *T*)*_il_* is the fixed interaction effect of diet *i* with the collection time *l*, and *E_ijkl_* ~*N*(0, *σ*^2^*_e_*) represents the residual error.

Orthogonal polynomial contrasts were used to determine linear and quadratic effects. Differences were declared statistically significant at *p <* 0.05. When there were significant interactions, the PLM procedure of SAS and the Tukey–Kramer test were used.

For the variables with a quadratic response, curve fitting and equations were obtained by regression analysis. Maximum or minimum SC+UAS inclusion levels were obtained setting the first derivative of the equation (*y = ax^2^ + bx + c*) equal to zero. Thus, the respective maximum or minimum response values for the parameters studied were estimated by replacing “*x*” in the equation with the SC+UAS inclusion level previously estimated.

## 3. Results

### 3.1. DM, DOM, and iNDF Intake

The intake of DM (g/d), DOM, and DM, expressed as g/kg BW, showed a quadratic response (*p <* 0.05; Table 3), with maximum values of 1281 g/d, 910 g/d, and 27 g/kg BW, respectively, for the same inclusion levels of 356, 414, and 343 g SC+UAS/kg DM. On the other hand, iNDF intake decreased linearly with increasing SC+UAS inclusion (*p* < 0.001).

### 3.2. Ruminal Fermentation

Rumen pH and RAN (Table 4) showed linear reductions when the diet included increasing levels of SC+UAS (*p <* 0.001). There was a quadratic response for both pH and RAN as a function of collection time (*p <* 0.001). A minimum pH value (6.18) was measured 5.17 h after the morning feeding, whereas a maximum RAN value (188.8 mg/L) was detected 1.84 h after feeding.

No interaction was seen between inclusion levels of SC and collection time regarding pH (*p =* 0.354), whereas RAN did exhibit an interaction (*p <* 0.001; Table 4 and Figure 1). At time 0 (before feeding), the lowest concentration was observed at 600 g SC+UAS/kg DM (68 mg/L). All treatments displayed greater RAN concentrations at 2 h (*p <* 0.001), with the highest concentration detected in the 600 g SC+UAS/kg DM treatment (334.2 mg/L; *p <* 0.001). Four hours after feeding, the highest RAN concentration was observed for 0 g SC+UAS/kg DM (167.1 mg/L), and the lowest for 300 g SC+UAS/kg DM (*p =* 0.022). The 0 g SC+UAS/kg DM treatment led to similar RAN concentrations at 0, 4, 6 and 8 h, and the same behavior was observed at the level of 150 g SC+UAS/kg DM.

Acetate and propionate concentrations (Table 4) increased linearly with increasing SC+UAS inclusion (*p <* 0.05). Collection time showed a quadratic response (*p <* 0.001). Maximum concentrations were estimated at 61.7 and 21.6 mmol/L, at 4.77 and 5.17 h after feeding, respectively, for the same acid order (Table 4). There was an interaction between inclusion level and collection time for propionate concentration (*p =* 0.033; Table 4 and Figure 2). Analyzing each inclusion level during 8 h of collection, the levels of 0 and 150 g SC+UAS/kg DM did not differ in acid concentration for the five collection times (*p =* 0.896 and 0.471, respectively). Other inclusion levels led to changes in propionate concentration (*p <* 0.01), with lower values at the first collection (0 h). The level of 300 g SC+UAS/kg DM increased propionate concentrations at each collection time, while for 450 and 600 g SC+UAS/kg DM, higher concentrations were detected at 4 and 6 h, respectively. Analyzing the response of the inclusion levels of SC at each collection time, the greatest overall propionate concentrations were observed at 600 g SC+UAS/kg DM, except at 2 h when similar concentrations were observed for 450 and 600 g SC+UAS/kg DM. Lower propionate concentrations were observed at the two lower SC inclusion levels (0 and 150 g SC+UAS/kg DM).

Butyrate concentration (Table 4) was not influenced by inclusion of SC or collection time, nor was there an interaction between inclusion level and collection time (*p >* 0.05). Acetate/propionate ratio (Table 4) decreased linearly with increasing SC+UAS inclusion (*p <* 0.001). A quadratic decrease (*p <* 0.001) was observed in the acetate/propionate ratio with collection time, with a minimum value of 3.17 detected 5.70 h after feeding. Total VFA concentration increased linearly with increasing SC+UAS inclusion (*p =* 0.001; Table 4). Regarding collection time, total VFA concentration showed a quadratic response (*p <* 0.001), with a maximum concentration of 93.6 mmol/L detected at 4.94 h after the first feeding.

### 3.3. Nitrogen Balance, Nitrogen Compounds, and Blood Glucose

Nitrogen ingested (g/d and g/kg BW^0.75^) and fecal N (g/d) showed a quadratic increase (*p <* 0.05; Table 5), with maximum values of 30 g/d, 1.66 g/kg BW^0.75^ and 6.45 g/d, respectively, for the same level order of 363.3, 300, and 244.4 g SC+US/kg DM. Nitrogen excretion via feces (g/kg BW^0.75^ and g/kg N consumed) and N absorbed (g/d and g/kg BW^0.75^) were not affected by inclusion (*p >* 0.05).

Nitrogen balance is presented in Table 5. Urinary N-excretion (g/d; g/kg BW^0.75^) and the N-intake to DOM-intake ratio decreased linearly (*p <* 0.05) with increasing SC+UAS inclusion. A quadratic response (*p =* 0.019) was observed in urinary N-excretion (g/kg N-intake), with a minimum excretion of 332.2 g/kg N detected at 393.1 g SC+UAS/kg DM. Nitrogen retained showed a quadratic response (*p <* 0.05) with SC+UAS inclusion. Maximum N-retention was estimated at 14.1 g/d, 0.72 g/kg BW^0.75^, and 593.1 g N/kg N absorbed, measured at levels 438.3, 350 g, and 440.1 SC+UAS/kg DM, respectively. Nitrogen retained decreased linearly (*p =* 0.008) when considering the N ingested, whereas N-retention considering the N absorbed showed a quadratic response (*p =* 0.044). Maximum retention was estimated at 593.1 g N/kg N absorbed at the level of 440.1 g SC+UAS/kg DM.

Concentrations of serum urea, plasma ammonia, urine ammonia and N-excretion decreased as urine ammonia (g/d) decreased, while serum glucose increased linearly (*p <* 0.05) with increasing SC+UAS inclusion (Table 5). Urinary urea concentration, daily excretion of N-urea and N excreted as ammonia expressed as a percentage of total N excreted in the urine were not influenced by inclusion (*p >* 0.05). However, N-excretion as urea, expressed as a percentage of N excreted via urine, displayed a quadratic response (*p =* 0.038) with a maximum value 76.8% at 538 g DM/kg SC+UAS.

### 3.4. Urinary Volume, PD, and MPS

Urinary volume and urine allantoin concentration were similar (*p >* 0.05), while daily uric acid concentration and excretion increased linearly with increasing SC+UAS inclusion (*p <* 0.05; Table 6). Allantoin daily excretion, excreted and absorbed PD, MNS and MPS showed quadratic responses (*p <* 0.05) with maximum values estimated at 8.83 mmol/d, 8.51 mmol/d, 10.9 mmol/d, 7.08 g/d N, and 48.8 g/d MPS, respectively, for the same order of SC+UAS inclusion levels of 585, 345, 462.5, 335, and 418 g/kg DM. The efficiency of MPS per kg of DOM intake or kg of DOMR intake was not influenced by inclusion of SC+UAS (*p >* 0.05).

## 4. Discussion

### 4.1. DM, DOM, and iNDF Intake

Dry matter intake (Table 3) increased quadratically from level 0 to 355.43 g SC+UAS/kg DM, probably due to the inclusion of SC, which has a highly effective ruminal DM degradability (711 g/kg DM) [8] as a result of its low NDF and lignin concentrations (148 and 10 g/kg DM, respectively), and its high NFC concentration (712 g/kg DM; Table 1). In addition, the SC+UAS inclusion resulted in decreased iNDFI to levels below 150 g/kg DM, a value above which DM intake can be limited [31].

Reduction in DM and DOM intakes (Table 3) were possibly due to metabolic regulation as a consequence of the increased propionate concentrations observed in this experiment (Table 4 and Figure 2). Farningham and Whyte [32] reported that propionate flow on the portal system has an important role in regulating intake, which is independent of changes regarding plasma insulin level. Oba and Allen [33] observed that increasing propionate concentrations through ruminal infusions decreased DM intake, metabolizable energy intake, meal size, and meal frequency. Propionate uptake by the liver can be used for gluconeogenesis, utilization of ATP, or oxidation in the tricarboxylic acid cycle through acetyl CoA. Propionate uptake during meals stimulates oxidation of acetyl CoA to CO_2_, rapidly generating ATP and stimulating satiety [34].

### 4.2. Ruminal Fermentation

Quadratic decrease in pH with collection time (Table 4) was caused by dietary carbohydrate fermentation [35], which generated the same quadratic response in total VFA concentration (Table 4). Subsequently, pH increased gradually as a consequence of rumen acids being extracted either via liquid phase passage or by absorption through the rumen wall. The linear reduction in rumen pH with increasing SC+UAS inclusion can be an effect of the fermentation of the high NFC levels in the SC diets (Table 2), resulting in greater total VFA production. Moreover, SC inclusion diminished rumination [7] due to low NDF concentration, which reduces saliva secretion and, consequently, rumen buffering capacity [36]. Ruminal pH did not reach values lower than 6.0 with SC+UAS inclusion. Therefore, cellulolysis [37,38] and NDF digestibility could not be affected, as reported by Siqueira et al. [9].

The greatest RAN concentration was seen at 600 g SC+UAS/kg DM two hours after feeding, despite showing the lowest RAN concentration at zero hours (Figure 1), possibly due to the greater UAS proportion and its high solubility in the rumen. The greater reduction at four hours with 300–600 g SC+UAS/kg DM was possibly due to greater ruminal synchrony of N to readily fermentable carbohydrate [39], which could be a consequence of a greater NFC content of SC, plus the use of two rapidly rumen-degradable non-protein N-sources, such as UAS. The use of that source in diets with rapidly rumen-fermentable energy promotes greater RAN efficiency by rumen microbes in microbial protein synthesis [40,41].

Ruminal pH reduction resulting from SC+UAS inclusion could influence the lower RAN concentration from 300 g SC+UAS/kg DM level after the peak at two hours following the first feeding (Figure 1). The ruminal wall permeability toward ammonia absorption as NH_3_ is 175 times greater than as NH_4_^+^ when the ruminal pH is below 6.4. However, NH_3_ concentration is lower than NH_4_^+^; therefore, 70% of the ammonia will be absorbed through the rumen wall as NH_4_^+^ by potassium-facilitated transport [42]. Thus, NH_4_^+^ ions cannot rapidly be removed through the rumen wall into the blood [43], meaning that they are stored for longer in the rumen. Therefore, NH_4_^+^ ions can be captured by microbes for protein synthesis [44], consequently decreasing RAN.

Inclusion levels of SC at all collection times promoted RAN concentrations greater than 50 mg/L (Figure 1). According to Satter and Slyter [45], this value is enough to support an adequate rumen bacterial growth rate. Two hours after feeding, all inclusion levels reached RAN concentrations greater than 235 mg/L, a value suggested by Mehrez et al. [46] to be the point of maximal dietary OM degradation. Detmann et al. [47] suggested that for diets based on medium- to-high-quality forages, supplementation should increase RAN concentrations above 160 mg/L, a value reached from 0–4 h at 0 g SC+UAS/kg DM.

Acetate concentration usually increases with structural carbohydrate fermentation in the diet [48]. However, in the present experiment there was a linear increase in acetate concentration with increasing SC+UAS inclusion (Table 4) as NDF decreased (Table 2), differing by 18% between the 0 and 600 g SC+UAS/kg DM treatments. The increase in the acetate concentration with SC inclusion could be due to its high content of fructans and *β*-glucans [49], both of which are constituents of the neutral detergent-soluble fiber [50]. Fructans are fermented by some rumen bacteria that produce acetate [51].

The increase of propionate concentration with increasing SC+UAS inclusion, with the highest values observed after feeding with 300 g SC+UAS/kg DM (Figure 2), could be a consequence of NFC fermentation [48]. Pectin is a constituent of NFC [50] and is found in SC [49]. It is rich in rhamnose [52], the anaerobic fermentation of which generates ethanol, acetone, and 1,2-propanediol [53]. The latter is fermented by rumen bacteria that produce propionate [54]. Greater propionic acid production with SC+UAS inclusion promotes a linear decrease in the ratio of acetate to propionate (Table 4), which is also related to rumen pH decrease [35].

### 4.3. Nitrogen Balance, Nitrogen Compounds, and Blood Glucose

The quadratic increase in N-intake (g/d and g/kg BW^0.75^) and its excretion through feces (Table 5) was possibly a consequence of the same response observed for DM intake (Table 3). Greater DM intake implies higher intake of concentrate consisting mainly of corn (Table 2). This ingredient increases the amount of microbial-originated N via feces, due to its greater fermentative activity in the large intestine [55].

Using diets higher in rumen-degradable protein increases the N-excretion via urine [56]. In this experiment, the inclusion of UAS to adjust the crude protein concentration of diets increased the amount of degradable N in the rumen, although urine N-excretion decreased (Table 5). The greatest N-intake was observed at 363 g SC+UAS/kg DM, although its excretion in feces decreased from 244 g SC+UAS/kg DM, in addition to the decreased urinary N-excretion. This response allows the inference of a greater N-use by animals at that level of SC+UAS inclusion in the diet. A greater use of energy and a protein diet for microbial protein synthesis generates lower N-loss [57], and the greater utilization was up to 438 g SC+UAS/kg DM when the greatest N-retention was reached.

Inclusion levels of 0 and 150 g SC+UAS/kg DM showed consumptions in excess of 33.6 g N/kg DOM; therefore, losses of protein or incomplete net transfer could occur [58], which explains the lower N-retention at these levels of inclusion. At 300 g SC+UAS/kg DM, consumption was higher than 16 g N/kg DOM; hence, there was no N-limitation for microbial protein synthesis [59]. However, retained N decreased from inclusion level of 438 g SC+UAS/kg DM; therefore, other factors generated the decrease in the use of N from this level of inclusion.

The dietary N-usage was also reflected on both serum urea and plasma ammonia (Table 5), with a linear decrease in their concentrations with increasing SC+UAS inclusion. Blood urea values were within the normal range for sheep (24–60 mg/dL) [60], and plasma ammonia concentrations were lower than the limit for intoxication (10–40 mg/L) [61].

Urine urea concentration and its daily excretion (Table 5) were not influenced by SC+UAS inclusion, although blood concentration levels differed. The amount of urea excreted in the urine is determined by the amount of urea filtered at the kidney glomeruli [62]. However, N-intake between 26.0 and 31.6 g/d (Table 5) could generate the same rate of clearance of urea from the plasma by the kidneys, as observed by Cocimano and Leng [63]; despite the greater concentration of urea in blood and similar amount in the urine (Table 6), a similar urea excretion was generated.

Szanyiová et al. [64] reported when sheep were fed high- or low-N diets (28.71 vs. 9.32 g/d), the N excreted through urine as urea as a proportion of total urinary N was not affected (74 vs. 69%). In the present experiment, N-excretion in the urine as urea as a proportion of the total urinary N showed a quadratic response, despite daily urea excretion in grams was similar between treatments; the difference is possibly due to the total N-excretion through urine being reduced.

Ammonia concentration and daily N-excretion as ammonia through urine (Table 5) showed the same response as serum urea and plasma ammonia. Free ammonia in extrahepatic tissues binds with glutamate to produce glutamine, which transports ammonia via blood to the liver to produce urea, or even to the kidneys, where glutamine degradation generates ammonia release in the urine [65]. Therefore, there was possibly a decrease in ammonia production by extrahepatic tissues with increasing SC+UAS inclusion.

Glucose concentrations were within the reference levels of 43–76 mg/dL, as suggested by Contreras et al. [60]. The linear increase in glucose concentration with increasing SC+UAS inclusion (Table 5) may be a consequence of the same response observed in rumen propionic acid concentration, since this acid is a substrate for gluconeogenesis and the main glucose source for ruminants [48,66]. Moreover, the dietary glucose absorption is greatly reduced in ruminants [67].

### 4.4. Urinary Volume, PD, and MPS

Vieira et al. [68] reported that the diuretic effect of SC is promoted by its high K and low Na contents, despite this inclusion having no influence on urinary volume (Table 6). This response was different from those reported by other authors when SC was included in the diet [69,70].

The quadratic increase in PD excretion with increasing SC+UAS inclusion (Table 6) could be a consequence of the same effect, as observed for DM intake, given the correlation between them when DM intake is adjusted for BW [16,71]. For example, greater PD excretion was observed at 345 g SC+UAS/kg DM, which matched the greater DM intake (g/kg BW) at 343 g SC+UAS/kg DM.

According to Van Soest [36], greater DM intake reduces the energetic cost of maintaining rumen microbes because the residence time in the rumen is also decreased. In addition, due to the greater DM intake, rumen particle flow is increased and the number of bacteria adhering to feed that escapes from rumen to abomasum and duodenum is increased, so the flow of microbial N is greater, promoting higher microbial purine absorption [15]. At 335 g SC+UAS/kg DM, the maximum MNS was produced, at which the RAN concentration was estimated at 147 mg/L (RAN = 164.96 − 0.0537 × SC + UAS), close to that observed by Detmann et al. [72], who reported maximum MNS at a RAN concentration of 145 mg/L.

In the present experiment, a quadratic increase in MPS was observed. In contrast, Cardoso et al. [5] and Barros et al. [73] found a linear increase in MPS when SC was included in lamb (450 g/kg DM) and heifer (500 g/kg DM) diets, respectively. This different response may be due to the roughage/concentrate ratio being 50:50 in the experiments mentioned above, and 70:30 in our research.

The gradual increase of MPS until the inclusion reached 418 g SC+UAS/kg DM could be due to the use of the energy contained in the NFC of the SC, mostly as a result of rapid ruminal degradation [49], and the energy provided by slow degradation of the structural carbohydrates of Tifton hay (cellulose and hemicellulose). These two different rates of carbohydrate degradation are able to maintain a constant supply of ATP for microbial growth [74]. Additionally, the feed contained non-protein N-sources for rapid ruminal degradation (a UAS mix) that maximize microbial growth, and bacteria that ferment NFC grow faster than those that ferment structural carbohydrates [75,76].

In addition to the DM intake reduction from the level 356 g DM/kg, which could negatively influence microbial protein synthesis, the decrease in rumen pH (close to 6.0) would contribute to lower MPS, even when a reduction in RAN with SC+UAS inclusion is associated with greater microbial protein utilization. Ruminal pH values of 6.0 generate lower ATP synthesis because bacteria use it for non-growth functions, such as intracellular pH maintenance [77].

With decreases in RAN, blood urea and ammonia concentrations (as indicators of dietary N-utilization), and consumption over 16 g N/kg DOM, the lower MPS as from 418 g SC+UAS/kg DM would be directly associated with the rumen environment. The great amounts of fermentable carbohydrates in diets with high SC+UAS levels increase VFAs and consequently decrease rumen pH, which increases bacterial energy demand. This lower production of MPS could explain the decrease in retained N (g/d).

Despite SC+UAS inclusion having an influence on MNS and MPS, the efficiency of MPS was not influenced by inclusion (Table 6). The values of the efficiency of MPS per kg of DOM intake and kg of DOMR were lower than those recommended by Balch [78] (130 g MPS/kg DOM, or 200 g MPS/kg DOMR). Pereira et al. [79] also observed low MPS efficiency (47.4 g/kg total digestible nutrient intake) in sheep fed ad libitum under tropical conditions, a value lower than that recommended by NRC [14] (130 g/kg).

## 5. Conclusions

In diets for wethers with a roughage/concentrate ratio of 70:30, roughage consisting of a SC+UAS/hay (Tifton-85) ratio of 41:29 is recommended to maximize DOM intake, N-retention, and MPS. This could lead to improvements in the productive performance of the animals in semi-arid regions.

## Figures and Tables

**Figure 1 animals-12-00401-f001:**
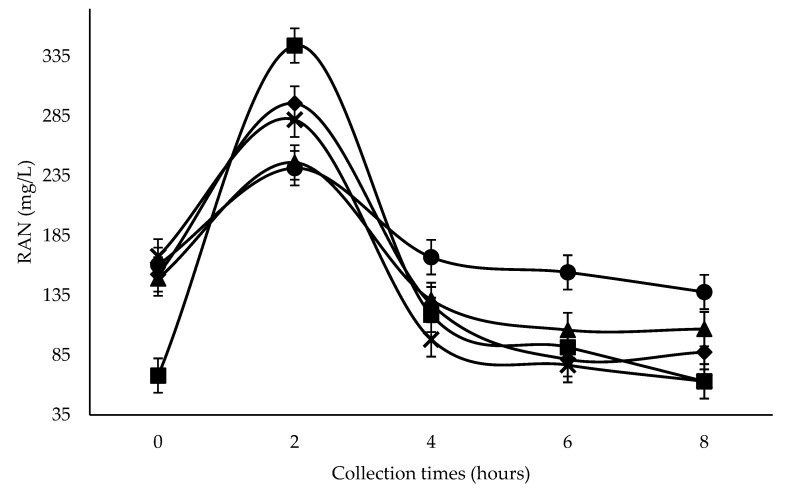
Effect of Treatment × Collection time interaction (*p* < 0.001) on rumen ammonia nitrogen (RAN) concentration in wethers feeding with 0 (●), 150 (▲), 300 (♦), 450 (**×**) and 600 (■) g SC+UAS/kg DM.

**Figure 2 animals-12-00401-f002:**
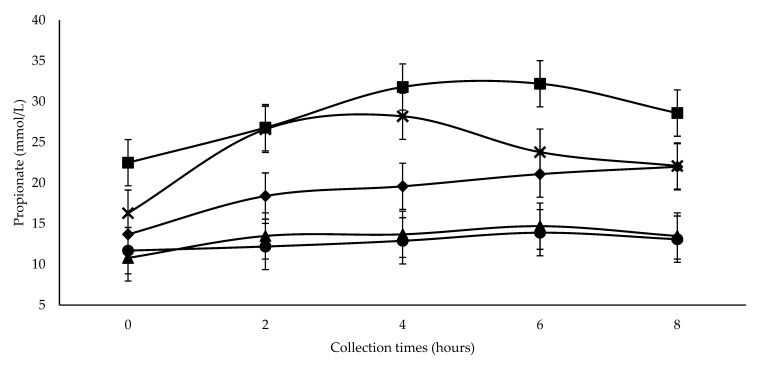
Effect of Treatment × Collection time interaction (*p* = 0.033) on propionate concentration in wethers feeding with 0 (●), 150 (▲), 300 (♦), 450 (**×**) and 600 (■) g SC+UAS/kg DM.

**Table 1 animals-12-00401-t001:** Chemical composition of ingredients (g/kg DM).

Item	Tifton-85 Hay	Spineless Cactus	Soybean Meal	Ground Corn	Urea	Ammonium Sulfate
DM ^a^	929	223	918	899	991	994
Ash	103	92	74	16	-	-
OM	897	908	926	984	-	-
CP	118	33	518	100	2900	1295
EE	14	15	16	44	-	-
aNDF(n)	651	148	97	80	-	-
iNDF	298	80	1.6	4.0	-	-
ADF	366	98	85	30	-	-
Lignin	62	10	4.3	10	-	-
NFC	113	712	295	761	-	-
TC	765	860	392	841	-	-

^a^ g DM/kg of fresh weight. DM: dry matter; OM: organic matter, CP: crude protein; EE: ether extract; aNDF(n): neutral detergent fiber corrected for ash and nitrogenous compounds; iNDF: indigestible neutral detergent fiber; ADF: acid detergent fiber; NFC: non-fibrous carbohydrates; TC: total carbohydrates.

**Table 2 animals-12-00401-t002:** Proportion of ingredients and chemical composition of the experimental diets.

Ingredients (g/kg DM)	Inclusion of SC+UAS (g/kg DM)				
	0	150	300	450	600
Tifton-85 hay	693	549	399	250	99
Spineless cactus	0	147	293	437	583
Ground corn	208	204	203	204	205
Soybean meal	84	81	82	82	82
Urea + Ammonium sulfate ^a^	0	4	8	12	16
Common salt	5	5	5	5	5
Mineral mix ^b^	10	10	10	10	10
Chemical composition (g/kg DM)					
DM ^c^	922	629	478	387	325
OM	919	921	923	925	927
CP	146	143	140	141	140
EE	20	20	20	21	21
aNDF(n)	476	404	327	252	175
iNDF	208	176	143	111	77
ADF	267	229	188	148	107
NFC	277	361	449	534	621
TC	753	757	762	765	769
aNDF(n) from Tifton-85 hay	452	358	260	163	65

^a^ 9 parts of urea and 1 part of ammonium sulphate. ^b^ Assurance levels provided by the manufacturer: (g/kg) 120 Ca, 87 P, 147 Na, 18 S, (mg/kg) 590 Cu, 40 Co, 20 Cr, 1800 Fe, 80 I, 1300 Mn, 15 Se, 3800 Zn, 10 Mo and 870 F (maximum). ^c^ g DM/kg of fresh weight. DM: dry matter; OM: organic matter; CP: crude protein; EE: ether extract; aNDF(n): neutral detergent fiber corrected for ash and nitrogenous compounds; iNDF: indigestible neutral detergent fiber; ADF: acid detergent fiber; NFC: non-fibrous carbohydrates; TC: total carbohydrates.

**Table 3 animals-12-00401-t003:** Effect of SC+UAS inclusion on dry matter, digestible organic matter, and indigestible neutral detergent fiber intakes.

Item	Inclusion of SC+UAS (g/kg DM)					SEM	*p*-Value	
	0	150	300	450	600		L	Q
DMI								
g/d	1110	1179	1379	1185	1236	78.6	0.066	0.014 ^a^
g/kg BW	24	26	30	26	27	2.25	0.086	0.027 ^b^
DOMI (g/d)	692	782	984	839	905	59.0	0.001	0.012 ^c^
iNDFI								
g/d	217	189	171	106	91	15.5	0.000	0.577
g/kg DMI	194	160	124	89	73	5.29	0.000	0.175

DMI: dry matter intake; BW: body weight; DOMI: digestible organic matter intake; iNDFI: indigestible neutral detergent fiber intake; SEM: standard error of the mean; L: linear; Q: quadratic. ^a^ DMI (g/d) = −0.0014(SC+UAS)^2^ + 0.9952(SC+UAS) + 1104.5. ^b^ DMI (g/kg BW) = −0.00003(SC+UAS)^2^ + 0.0206(SC+UAS) + 23.869. ^c^ DOMI = −0.0013(SC+UAS)^2^ + 1.0759(SC+UAS) + 687.21.

**Table 4 animals-12-00401-t004:** Effects of SC+UAS inclusion on ruminal fermentation of wethers.

Item	Inclusion of SC+UAS (g/kg DM)					SEM	Collection Times (Hours)					SEM	*p*-Value				
	0	150	300	450	600		0	2	4	6	8		SC+UAS		Collection Time		Interaction
													L	Q	L	Q	
pH	6.54	6.47	6.27	6.22	6.16	0.07	6.62	6.33	6.20	6.21	6.30	0.05	<0.001	0.495	<0.001	<0.001 ^a^	0.354
RAN ^1^	172	148	149	138	137	8.10	140	281	129	102	92	6.44	0.005	0.258	<0.001	<0.001 ^b^	< 0.001
Acetate ^2^	53	54	59	59	63	4.46	51	59	61	61	57	3.41	0.048	0.978	0.019	<0.001 ^c^	0.777
Propionate ^2^	13	13	19	23	28	2.50	15	20	21	21	20	1.54	<0.001	0.386	<0.001	<0.001 ^d^	0.033
Butyrate ^2^	9	9	10	10	12	0.97	10	10	10	11	10	0.53	0.074	0.798	0.108	0.214	0.655
A:P ^3^	4.2	4.1	3.2	2.8	2.3	0.26	3.7	3.4	3.2	3.2	3.3	0.14	<0.001	0.817	<0.001	<0.001 ^e^	0.146
Total VFA ^2^	75	77	88	93	103	6.78	75	88	93	92	87	5.00	0.001	0.681	0.001	<0.001 ^f^	0.494

RAN: rumen ammonia nitrogen. ^1^ mg/L. ^2^ mmol/L. ^3^ Acetate:Propionate ratio. SEM: standard error of the mean; L: linear; Q: quadratic. Linear and quadratic effects are significant at *p* < 0.05. H: hour. ^a^ pH = 0.0161H^2^ − 0.1666H + 6.6126. ^b^ RAN = −3.1732H^2^ + 11.651H + 178.15. ^c^ Acetate= −0.4732H^2^ + 4.5107H + 50.914. ^d^ Propionate = −0.2393H^2^ + 2.4743H + 15.166. ^e^ A:P: 0.0155H^2^ − 0.1768H + 3.6763. ^f^ VFA = −0.7429H^2^ + 7.3329H + 75.497.

**Table 5 animals-12-00401-t005:** Apparent nitrogen balance, serum and urine urea, plasma and urine ammonia, and serum glucose.

Item	Inclusion of SC+UAS (g/kg DM)					SEM	*p*-Value	
	0	150	300	450	600		L	Q
N intake								
g/d	26.0	28.1	31.6	26.9	27.8	1.99	0.485	0.024 ^a^
g/kg BW^0.75^	1.46	1.59	1.76	1.51	1.56	0.12	0.503	0.022 ^b^
N fecal								
g/d	5.86	6.42	6.61	5.78	5.42	0.45	0.153	0.033 ^c^
g/kg BW^0.75^	0.33	0.36	0.37	0.33	0.31	0.03	0.191	0.053
g/kg N intake	233	231	209	216	194	13.5	0.055	0.832
N urinary								
g/d	11.8	10.7	10.1	9.46	9.71	0.65	0.011	0.214
g/kg BW^0.75^	0.66	0.60	0.56	0.53	0.54	0.04	0.011	0.238
g/kg N intake	475	383	318	356	349	29.1	0.006	0.019 ^d^
N absorbed								
g/d	20.1	21.6	25.0	21.1	22.4	1.71	0293	0.097
g/kg BW^0.75^	1.13	1.22	1.40	1.18	1.26	0.10	0.278	0.083
N retained								
g/d	8.31	11.0	14.9	11.7	12.7	1.44	0.021	0.028 ^e^
g/kg BW^0.75^	0.47	0.62	0.84	0.65	0.72	0.09	0.012	0.019 ^f^
g/kg N intake	292	387	473	428	457	38.2	0.008	0.081
g/kg N absorbed	374	502	597	544	567	45.4	0.008	0.044 ^g^
g N/kg DOMI	37.7	36.0	32.2	32.1	30.8	0.89	0.000	0.183
Serum urea (mg/dL)	54.1	53.8	47.1	43.4	43.8	3.58	0.003	0.663
Urine urea (mg/dL)	873	744	1081	1036	858	103	0.438	0.237
Urea N (g/d)	6.05	5.85	7.33	7.01	6.37	0.83	0.440	0.325
% of urine N	51.1	53.8	71.5	74.7	64.9	5.48	0.005	0.038 ^h^
Plasma ammonia (mg/L)	6.30	6.35	5.95	4.88	5.04	0.48	0.004	0.745
Urine ammonia (mg/L)	185	89	139	121	93	29.1	0.031	0.373
Ammonia N (g/d)	0.22	0.12	0.14	0.13	0.11	0.04	0.034	0.187
% of urine N	1.85	1.16	1.36	1.35	1.14	0.34	0.167	0.458
Glucose (mg/dL)	50.1	54.9	55.7	59.7	56.1	3.97	0.007	0.053

N: nitrogen; BW: body weight; DOMI: digestible organic matter intake; SEM: standard error of the mean; L: linear; Q: quadratic. Linear and quadratic effects are significant at *p* < 0.05. ^a^ N intake (g/d) = −0.00003(SC+UAS)^2^ + 0.0218(SC+UAS) + 26.086. ^b^ N intake (g/kg BW^0.75^) = −0.000002(SC+UAS)^2^ + 0.0012(SC+UAS) + 1.4691. ^c^ N faecal = −0.000009(SC+UAS)^2^ + 0.0044(SC+UAS) + 5.9134. ^d^ N urinary = 0.0009(SC+UAS)^2^ − 0.7076(SC+UAS) + 471.31. ^e^ N retained (g/d) = −0.00003(SC+UAS)^2^ + 0.0263(SC+UAS) + 8.3289. ^f^ N retained (g/kg BW^0.75^) = −0.000002(SC+UAS)^2^ + 0.0014(SC+UAS) + 0.4726. ^g^ N retained (g/kg N absorbed) = −0.0011(SC+UAS)^2^ + 0.9682(SC+UAS) + 380.01. ^h^ Urea N (%) = −0.0001(SC+UAS)^2^ + 0.1076(SC+UAS) + 47.857.

**Table 6 animals-12-00401-t006:** Urinary volume, purine derivatives excretion and absorption, and microbial nitrogen and protein supply.

Variable	Inclusion of SC+UAS (g/kg DM)					SEM	*p*-Value	
	0	150	300	450	600		L	Q
Urinary volume (L/d)	1.52	1.79	1.51	1.46	1.61	0.20	0.751	0.987
Allantoin (mmol/L)	3.86	4.11	6.49	5.18	5.03	0.77	0.148	0.111
Allantoin (mmol/d)	5.45	6.49	8.46	7.03	7.51	0.57	0.006	0.019 ^a^
Uric acid (mmol/L)	0.50	0.60	0.97	0.89	0.93	0.10	0.002	0.132
Uric acid (mmol/d)	0.73	0.95	1.35	1.16	1.40	0.11	0.000	0.116
PD excreted (mmol/d)	6.18	7.43	9.81	8.19	8.91	0.60	0.001	0.011 ^b^
PD absorbed (mmol/d)	6.71	8.43	11.5	9.44	10.3	0.79	0.001	0.009 ^c^
MNS (g/d)	4.87	6.12	8.35	6.86	7.49	0.58	0.001	0.009 ^d^
MPS (g/d)	30.5	38.3	52.2	42.9	46.9	3.60	0.001	0.009 ^e^
g MPS/kg DOMI	44.4	49.8	53.2	51.2	51.6	3.57	0.146	0.227
g MPS/kg DOMR	68.3	76.6	81.8	78.7	79.4	5.49	0.146	0.225

PD: purine derivatives; MNS: microbial nitrogen supply; MPS: microbial protein supply; DOMI: digestible organic matter intake; DOMR: digestible organic matter apparently digested in the rumen; SEM: standard error of the mean; L: linear; Q: quadratic. Linear and quadratic effects are significant at *p* < 0.05. ^a^ Allantoin (mmol/d) = −0.00001(SC+UAS)^2^ + 0.0117(SC+UAS) + 5.4103. ^b^ PD excreted (mmol/d) = −0.00002(SC+UAS)^2^ + 0.0138(SC+UAS) + 6.1371. ^c^ PD absorbed (mmol/d) = −0.00002(SC+UAS)^2^ + 0.0185(SC+UAS) + 6.6594. ^d^ MNS (g/d) = −0.00002(SC+UAS)^2^ + 0.0134(SC+UAS) + 4.8334. ^e^ MPS (g/d) = −0.0001(SC+UAS)^2^ + 0.0836(SC+UAS) + 30.28.

## Data Availability

Data are available upon request from the corresponding authors.

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
