# Peer review of "Spineless Cactus plus Urea and Tifton-85 Hay: Maximizing the Digestible Organic Matter Intake, Ruminal Fermentation and Nitrogen Utilization of Wethers in Semi-Arid Regions"

_animals, 2022, doi:10.3390/ani12030401_

Round 1

Reviewer 1 Report

This is a study of the effects of replacing increasing amounts of Tifton-85 hay with spineless cactus supplemented with urea and ammonia sulfate on the metabolism of wether sheep. In almost all respects, the study appeared to be conducts with adequate scientific controls and techniques.  There were several areas where the writing needs to be improved.

Major difficulties:

Several areas you indicate that x increased (or decreased) when SC-UAS was fed, please make sure you state, that this is with increasing amounts of SC-UAS.

Where were the Tifton hay and spineless cactus sourced? On the experiment station? The local market? Far away? Where?

Lines 223-225. What technique was used to calculate these numbers? You need to describe this somewhere.

Lines 245-248. Same comment as above.

Lines 264-267. Something is missing here. I do not understand what you are trying to say.

Line 274. Again, I think you need to clarify this. I suggest going with the same nomenclature as you used for the treatments. I can generally follow what you mean, but not this time.  Try to make it easy on the reader.

Minor English

Line 88. "Drinkers" is a term used for alcoholics. 'Waterers' would be the preferred term.

Table 2

Line aNDF(n) by Tifton-85 hay. I suggest replacing "by" with 'from'.  It is better in English

Line 135. Technically "genital organs" are not present in wethers. Please just call it the 'penis' and be done with it.

Line 158-159. These phrases are a bit difficult for English speaking people.  I suggest, 'Acid detergent fiber and lignin for all ingredients were determined according to Van Soest [22] with the exception that lignin concentration of SC was determined by potassium permanganate oxidation [23].

Line 191. Isn't this an average of one study?  Doesn't the digestibility of OM depend on the lignin %, NDF% and NFC%? Think you want to rephrase this so that the phrase does not raise red flags. I might suggest, the '..was calculated assuming 65% of DOM intake []'

Line 213. Please make sure the labels in the tables are the same as in the text. For the different DM calculations, it is a bit confusing.

Line 489. Suggest you place a 'to' after "lead".

Author Response

REVIEWER 1

This is a study of the effects of replacing increasing amounts of Tifton-85 hay with spineless cactus supplemented with urea and ammonia sulfate on the metabolism of wether sheep. In almost all respects, the study appeared to be conducts with adequate scientific controls and techniques.  There were several areas where the writing needs to be improved.

Major difficulties:

  1. Several areas you indicate that x increased (or decreased) when SC-UAS was fed, please make sure you state, that this is with increasing amounts of SC-UAS.

Response: It was attended. We hope it was what you have asked for.

  1. Where were the Tifton hay and spineless cactus sourced? On the experiment station? The local market? Far away? Where?

Response: It was attended. All the hay to be used during the experiment was purchased at once, to avoid changes in its chemical composition, in case another batch was purchased.

  1. Lines 223-225. What technique was used to calculate these numbers? You need to describe this somewhere.

  1. Lines 245-248. Same comment as above.

  1. Lines 264-267. Something is missing here. I do not understand what you are trying to say.

  1. Line 274. Again, I think you need to clarify this. I suggest going with the same nomenclature as you used for the treatments. I can generally follow what you mean, but not this time.  Try to make it easy on the reader.

3, 4, 5, and 6. They were attended. L220-225.

“For the variables with a quadratic response, curve fitting and equation were obtained by regression analysis. Maximum or minimum response values for the variables studied were obtained by equation derivation (ax2 + bx + c = 0). Thus, the respective SC+UAS inclusion level was estimated by replacing x with the maximum or minimum value in the equation.”

Minor English

  1. Line 88. "Drinkers" is a term used for alcoholics. 'Waterers' would be the preferred term.

Response: It was attended.

  1. Table 2. Line aNDF(n) by Tifton-85 hay. I suggest replacing "by" with 'from'.  It is better in English

Response: It was attended.

  1. Line 135. Technically "genital organs" are not present in wethers. Please just call it the 'penis' and be done with it.

Response: It was attended.

  1. Line 158-159. These phrases are a bit difficult for English speaking people. I suggest, 'Acid detergent fiber and lignin for all ingredients were determined according to Van Soest [22] with the exception that lignin concentration of SC was determined by potassium permanganate oxidation [23].

Response: It was attended.

  1. Line 191. Isn't this an average of one study?  Doesn't the digestibility of OM depend on the lignin %, NDF% and NFC%? Think you want to rephrase this so that the phrase does not raise red flags. I might suggest, the '..was calculated assuming 65% of DOM intake []'

Response: It was attended.

  1. Line 213. Please make sure the labels in the tables are the same as in the text. For the different DM calculations, it is a bit confusing.

Response: It was attended, Tables 1 and 2. It was clarified that DM in the chemical composition is expressed as g DM/ kg of fresh weight.

  1. Line 489. Suggest you place a 'to' after "lead".

Response: It was attended.

Reviewer 2 Report

  1. Maybe the “*” in author list for correspondence are missing.
  2. The DOI for references 7, 62, and 64 are missing.
  3. Please make sure of the difference between “–” and “-” and uniform them.

Please uniform the styles after carefully reading the "guide for author" and make carefully modification before uploading the revised manuscript.

Author Response

REVIEWER 2

  1. Maybe the “*” in author list for correspondence are missing.

Response: It was attended. The “*” was placed in the correspondence authors.

  1. The DOI for references 7, 62, and 64 are missing.

Response: It was attended the DOI for reference 62. References 7 and 64 have no DOI. The reference 64 just have PMDI.

  1. Please make sure of the difference between “–” and “-” and uniform them???

Response: Thank you for your observation. The differences in “–” and “-” were made by a native speaker who has proofread our paper. We assume the use of the “–” and “-” are corrected. But thank you for checking it. We will write to the Animals editor about it and see their suggestion. Again, thank you for your observations.
